# Role of the Ketogenic Diet Therapy and ACTH as Second Treatments in Drug-Resistant Infantile Epileptic Spasms Syndrome

**DOI:** 10.3390/nu17132085

**Published:** 2025-06-23

**Authors:** Anastasia Dressler, Letizia Bonfanti, Petra Trimmel-Schwahofer, Barbara Porsche, Simona Bertoli, Christoph Male

**Affiliations:** 1Department of Paediatrics and Adolescent Medicine, Comprehensive Center for Pediatrics, ERN EpiCARE, Medical University Vienna, A-1090 Wien, Austria; petra.trimmel.schwahofer@akhwien.at (P.T.-S.); barbara.porsche@akhwien.at (B.P.); christoph.male@meduniwien.ac.at (C.M.); 2Postgraduate School of Clinical Nutrition, University of Milan, 20122 Milan, Italy; letizia.bonfanti@unimi.it; 3IRCCS Auxologico Italiano, Research Laboratory on Nutrition and Obesity, 20145 Milan, Italy; simona.bertoli@unimi.it; 4Department of Food, Environmental and Nutrional Sciences (DeFENS), University of Milan, 20122 Milan, Italy

**Keywords:** infantile epileptic spasms syndrome, second treatment, ketogenic diet therapy, ACTH, anti-seizure medications

## Abstract

**Objectives**: The aim of this study was to evaluate the effectiveness of sequential treatments with adrenocorticotropic hormone (ACTH) or ketogenic diet therapy (KDT) in infants with infantile epileptic spasms syndrome (IESS) who did not achieve seizure freedom after initial treatment with either KDT or ACTH. **Methods**: We conducted a cohort study following a parallel-cohort randomized controlled trial comparing KDT with ACTH as first-line treatment for IESS. Infants who failed to respond were switched per protocol to the alternative treatment (ACTH or KDT) or, if this was not feasible or unsuccessful, to anti-seizure medications (ASMs). The primary outcome was the frequency of sustained seizure freedom at last follow-up. **Results**: Of 101 infants allocated to the initial treatment phase, N = 67 required further treatment. Of these, 31% (21/67) achieved sustained seizure freedom after the second treatment phase, and 15% (7/46) after rescue treatment with ASMs. KDT as the second treatment achieved sustained seizure freedom in 50% (12/24), compared to 19% (3/16) with ACTH and 9% (2/22) with ASMs. The cumulative response rate after the first and second treatments was 64% for the KDT-ACTH sequence and 68% for the ACTH-KDT sequence. The cumulative response after all three treatment phases was 78% (KDT-ACTH-ASM) and 72% (ACTH-KDT-ASM), respectively. **Conclusions**: KDT is at least as effective as ACTH as a second treatment and significantly more effective than ASMs in achieving sustained seizure freedom in infants with IESS.

## 1. Introduction

Infantile epileptic spasm syndrome (IESS) occurs in approximately 22 out of 100,000 live-born infants [1], accounting for approximately 10% of all epilepsies in the first years of life [2,3].

IESS was proposed by the International League Against Epilepsy (ILAE) as a new terminology to cover the clinical spectrum of all cases of epileptic spasms, even when hypsarrhythmia in the EEG and developmental delay are not or not yet present [4]. The syndrome may be preceded by Early Infantile Developmental Epileptic Encephalopathies (EIDEE), which exhibit the clinical phenotype of IESS at approximately 3 to 4 months of age [5]. Early seizure control and resolution of hypsarrhythmia are considered positive determinants of developmental outcome [6]; therefore, timely, effective, and safe treatment is essential. The longer hypsarrhythmia and epileptic spasms persist, the more delayed age-appropriate development becomes [7,8,9], and approximately 70% of children develop drug-resistant seizures with intellectual disability [10].

First-line therapies such as adrenocorticotropic hormone (ACTH), Vigabatrin (VGB), and oral steroids have proven effective [10,11,12,13,14], but have high relapse rates [15,16] and severe adverse effects with ACTH [12,17] and VGB [18,19,20]. Recently, the combination of ACTH and VGB has been reported to be more effective than hormonal treatment alone in terminating infantile epileptic spasms [9].

Ketogenic diet therapy (KDT) has shown substantial effectiveness [21,22,23,24,25,26,27,28,29,30,31] for refractory epileptic spasms after failure of standard treatment with hormonal therapy or VGB. Most studies were retrospective, but four prospective studies reported the use of KDT in IESS [23,29,30,31]. Hong and co-workers showed that 37% of infants became seizure-free, but 71% had been pre-treated with corticosteroids and VGB [23]. Our study group recently published the results of a prospective parallel cohort randomized controlled trial (PC-RCT) [29]. The study showed that KDT as an initial treatment was as effective as ACTH in achieving long-term seizure freedom but was better tolerated.

Sharma et al. reported the use of the Modified Atkins Diet (MAD) for the treatment of epileptic spasms refractory to hormonal treatment and randomized to either KDT or ASM. Their study showed that KDT was superior to the addition of ASM (23% vs. 0%) [30]. In a recent study, Schoeler et al. randomized infants under two years of age with drug-resistant epilepsy to either KDT or ASM and demonstrated comparable efficacy (seizure-freedom in 11% vs. 13%, respectively), as well as a better quality of life in the KDT group [31]. This cohort included 60% of refractory IESS.

In infants with IESS who do not respond to standard treatments, there is sparse evidence for optimal successive treatments [32]. Therefore, guidelines are primarily based on expert opinion [32,33]. In highly refractory IESS cases, response rates range from 9 to 44% [26,31,34], which is lower than with standard treatment, but higher than with some ASMs [14,30,31,32]. However, updated guidelines for the German-speaking countries and data from the KIWE trial support the use of KDT immediately after failure of hormonal treatments and VGB [31,32].

The present manuscript reports on the follow-up of our prospective PC-RCT in infants with IESS [29] who failed initial treatment with either KDT or ACTH. The aim of this study was to evaluate the effectiveness of successive treatments. Specific objectives were to assess sustained seizure freedom until the last follow-up visit:i.in response to ACTH or KDT as a second treatment;ii.in response to ASM as a second treatment (rescue treatment);iii.in response to ASM as third treatment (KDT-ACTH-ASM or ACTH-KDT-ASM); andiv.to determine the overall most effective treatment sequence.

Our hypothesis was that KDT is at least as effective as ACTH as a second treatment for refractory IESS.

## 2. Methods

### 2.1. Study Design

This is a cohort study reporting the follow-up of our prospective PC-RCT [29]. Infants failing to respond to the first treatment were switched to the respective alternative as prescribed by the study protocol. The study was performed at the Department of Pediatrics at the Medical University of Vienna and was approved by the institutional ethics committee (No. 542/2007).

### 2.2. Study Population

We included infants with (1) refractory IESS confirmed by Video-EEG-Monitoring after failure of the first treatment phase of the previously published PC-RCT [29] and (2) written informed consent of legal guardians. Figure 1 displays the flow of treatment sequences after the switch from the initial treatment allocation.

### 2.3. Study Treatments

KDT was gradually introduced at a 1:1 ratio (fat:non-fat ratio) without fasting and fluid restriction and individually increased to a maximum ratio of 3:1 (modified when beta-hydroxybutyrate reached levels >5 mmol/L). Daily energy and protein intake were age-appropriate according to nutritional guidelines [29]. ACTH was introduced according to the U.S. consensus report [12], as previously published [29]. Blood pressure was taken a minimum of six times daily, and laboratory tests (including inflammation parameters, blood count, electrolytes, and renal and liver function parameters) were performed a minimum of three times per week.

Before trial start (at baseline visit), all infants were fully evaluated, including a complete medical and metabolic work-up as well as neuroimaging [34]. Follow-up visits were scheduled weekly during the first month, and at 3 and 12 months. A final visit was scheduled at 24 months. Families kept a seizure diary to document seizure frequency, and 24 h Video-EEGs were performed at each visit to exclude epileptic spasms and/or hypsarrhythmia, using the definition published by Gibbs and Gibbs [35] and variants [36].

Two board-certified epileptologists independently assessed Video-EEGs and were blinded to treatment allocation and outcome. Inter-rater reliability was assessed prior to the study outcome. When raters disagreed, consensus was achieved by joint re-evaluation. Pediatric, nutritional, and neurological examinations [37,38,39], as well as Vineland Adaptive Behavior Scales II (VABS) [40], were performed at each visit. Treatment failure was defined as (1) not achieving electro-clinical remission (i.e., cessation of spasms and resolution of hypsarrhythmia) or (2) occurrence of relapse (i.e., recurrence of any seizure type) at any point before the last follow-up visit.

For the second treatment phase, the study protocol prescribed that infants be switched to the alternative treatment (ACTH or KDT, respectively) when the initial treatment was not successful. Infants in whom the second treatment as per protocol was not feasible or unsuccessful were switched to rescue treatment to ASMs or epilepsy surgery. Reasons for rescue treatment were as follows: clinical reasons, contraindications, or parents’ preference. Infants in evaluation for epilepsy surgery were scheduled for surgery as soon as possible.

As the study was observational in the second treatment phase, no randomization was performed. Treatment protocols and outcome assessments were identical for all treatment phases [29].

For the third treatment phase, further ASMs were administered as rescue treatment.

Treatment switches were performed as soon as possible according to guidelines [32,33].

Infants referred to us from other, non-tertiary centers often received VGB, which we continued as a co-medication and gave as monotherapy only for short periods, so that we did not consider VGB as a self-standing first treatment in the analysis, but created a subgroup analysis of treatment without or with VGB.

### 2.4. Study Outcome

The study outcome was sustained seizure freedom (i.e., from all seizure types, including epileptic spasms) until last follow-up. Outcome was assessed as follows:(1)in response to the second treatment (comparing KDT, ACTH, or ASMs);(2)response to rescue treatment (ASM);(3)cumulative response after first and second treatment phase (ACTH-KDT vs. KDT-ACTH); and(4)cumulative response after first, second, and successive treatment phases (ACTH-KDT-ASM, KDT-ACTH-ASM).

### 2.5. Data Analysis

For data analysis, we used IBM Statistical Package for Social Science (SPSS Statistics version 25). Descriptive statistics, medians, minimum, maximum values, absolute numbers, and percentages are reported. Frequencies of outcome were assessed separately for each treatment phase and cumulatively for successive treatment phases. For comparisons between treatment arms, odds ratios (OR) with 95% confidence intervals (95% CI) were calculated. As per protocol, infants were assigned to the second treatment phase (the respective other treatment); thus, randomization could not be feasible. We present unadjusted OR, reflecting a real-life cohort and the natural selection of patients due to the failure of prior treatment, and adjusted OR to account for imbalances in relevant co-variables.

The sample size for this follow-up cohort study was determined by the number of patients with failure of their prior treatment in the preceding study.

## 3. Results

From June 2008 to April 2017, 130 infants with infantile epileptic spasms syndrome (IESS) were screened; 29 did not fulfil the inclusion criteria, while 101 infants were enrolled, as previously described in detail [29]. Figure 1 shows the flow of participants from the initial treatment allocation through all successive treatments.

Furthermore, 96% of infants were still in follow-up after 6 months, 81% after 12 months, 65% after 24 months, and 39% after 48 months. The duration of follow-up was a median of 2 years (minimum 1 month; maximum 4 years). Time on treatment with KDT was a median of 1 year (minimum 1 month; maximum 6.1 years). Table 1 displays nutritional characteristics of the ketogenic diet prescription.

After initial KDT, 32 infants needed to be switched, and after initial ACTH, 35 infants needed to be switched (Figure 1). Patient characteristics per treatment arm of the second treatment phase are displayed in Table 2. Some differences were noted, such as a longer duration of epilepsy before trial start and a more frequent use of concomitant VGB in those receiving second KDT, reflecting a selection of highly refractory patients due to their failure of the first treatment. Appendix A shows baseline characteristics of patients who were switched after failure of initial treatment (either KDT or ACTH); no relevant differences were observed.

### 3.1. Study Outcomes

Table 3 shows the frequency of sustained seizure freedom until last follow-up per treatment phase (ACTH, KDT, ASM, epilepsy surgery)

#### 3.1.1. Second Treatment with KDT Versus ACTH

Of the 32 infants failing to respond to the first treatment with KDT, 16 were treated with a second ACTH, of whom 19% (3/16) showed sustained seizure freedom. Of the 35 infants failing first treatment with ACTH, 24 were treated with a second KDT, of whom 50% (12/24) showed a response (Table 3). The cumulative response after the first and second treatments was 24/37 (64%) with the KDT-ACTH sequence and 25/37 (68%) with the ACTH-KDT sequence (Table 4).

#### 3.1.2. Second Treatment with ASMs or Epilepsy Surgery as Rescue Treatment

Of the 67 infants failing to respond to the first treatment with either KDT or ACTH, 22 infants received a second treatment with ASMs, of whom 9% (2/22) showed sustained seizure freedom (Table 3; both after the sequence ACTH-ASM). Five infants received planned epilepsy surgery as a second treatment after prior KDT or ACTH, of whom 80% (4/5) showed sustained seizure freedom (three after initial KDT and 1 after initial ACTH).

#### 3.1.3. Response to Second Treatments Comparing Different Treatment Sequences

Table 5 shows a comparison of different treatment sequences, providing unadjusted odds ratios and odds ratios adjusted for gender, etiology, age, and the number of previously used ASMs. The sequence ACTH-KDT showed significantly higher probability of seizure freedom compared with KDT-ACTH or second ASM as rescue treatment. Surgery, however, showed the highest probability of seizure freedom (80%).

#### 3.1.4. Further Rescue Treatment with ASMs

Forty-six infants failing to respond to the first and second treatment phases received a further course of rescue treatment with ASMs. Of the 13 refractory infants with a prior KDT-ACTH sequence, 38% (5/13) showed sustained seizure freedom. Of the 12 refractory infants with a prior ACTH-KDT sequence, 17% (2/12) responded to further ASMs (Table 3). The cumulative response after first, second, and rescue treatments was 29/37 (78%) with the sequence KDT-ACTH-ASM and was 27/37 (72%) with the sequence ACTH-KDT-ASM (Table 4). Of the 21 infants who had failed the second treatment with ASMs or epilepsy surgery, none responded to further ASM treatment.

#### 3.1.5. Cumulative Response to First, Second, and Rescue Treatments as per Initial Treatment Allocation (Intention-To-Treat Analysis)

In all 53 infants who were initially allocated to KDT, the cumulative response after all treatment phases was 60% (32/53). In all 48 infants initially allocated to ACTH, the cumulative response after all treatment phases was 63% (30/48) (Table 4).

#### 3.1.6. Subgroup Analysis of Infants Without and With Prior and Concomitant Vigabatrin Treatment

Appendix A displays the frequency of sustained seizure freedom for the subgroups of infants without and with concomitant VGB per treatment phase.

In infants without prior VGB treatment, sustained seizure freedom was similar between arms in response to second treatment (KDT-ACTH 27%; ACTH-KDT 40%), and rescue treatment (KDT-ACTH-ASM 38%; ACTH-KDT-ASM 33%). In addition, cumulative response after first and second treatment (KDT-ACTH 62%; ACTH-KDT 75%), and cumulative response after all treatment phases (KDT-ACTH-ASM 76%; ACTH-KDT-ASM 83%) were largely similar (Appendix A).

In infants with concomitant VGB treatment (Appendix A), sustained seizure freedom in response to second treatment was more frequent for KDT (56%) compared to ACTH (ACTH 0%; risk difference 53% (95%CI 4–73%), *p* = 0.047). Response to rescue treatment with ASM tended to be higher after a KDT-ACTH-ASM sequence (40%) compared to an ACTH-KDT-ASM sequence (13%; *p* = 0.23). However, cumulative response after the first and second treatments (KDT-ACTH 69%; ACTH-KDT 67%) and after all treatment phases was largely similar between treatment arms (KDT-ACTH-ASM 81%; ACTH-KDT-ASM 71%) (Appendix A).

EEG data are displayed in Appendix A. Hypsarrhythmia was present in all patients, although discontinuous or incipient in n = 10 infants at baseline. Epileptic discharges were still present in half of the patients after 12 months, and sleep spindles were present in around two-thirds.

Adverse effects of KDT are shown in Appendix A, and growth data in Appendix A.

## 4. Discussion

We recently published a PC-RCT evaluating the initial treatment of infants with IESS, which demonstrated similar efficacy of KDT compared to ACTH in achieving sustained seizure freedom [29]. The present manuscript reports the prespecified observational follow-up of the previous study and aims to evaluate infants who did not respond to the first treatment and study the effectiveness of subsequent treatments with ACTH or KDT or, if not feasible, with ASMs. In the second treatment phase, infants who received KDT after initial ACTH treatment showed a higher frequency of sustained seizure freedom than infants who received a second ACTH after initial KDT treatment. Infants requiring rescue treatment with ASMs responded similarly after prior KDT-ACTH or ACTH-KDT sequence; the cumulative frequency of sustained seizure freedom after all treatment phases was also similar for both initial treatment allocations.

The main strength of this prospective study is that treatment allocation followed a rigorous study protocol, allowing for systematic comparison between KDT, ACTH, or ASMs during successive treatment phases. For the first treatment, allocation to KDT or ACTH was randomized or, if randomization was clinically impossible or consent could not be obtained, treatment followed the clinical criteria of standard-of-care (PC-RCT design) [29]. For the second treatment after failure of either KDT or ACTH, the study protocol provided for a switch to the other treatment (ACTH or KDT). If this was clinically impossible or not accepted by parents, infants were treated with other ASMs. A small subgroup of patients was eligible for epilepsy surgery.

Given the available options for the second treatment phase based on the existing literature [26,31,32,33,41,42], assignment to the other protocol treatment (ACTH or KDT) was expected to be superior to ASMs.

Randomization at this time point would not have been clinically and ethically appropriate, which is a limitation to the comparison of treatments. Differences in patient characteristics between these treatment arms, which may influence outcomes, result from selection due to failure of initial treatment and reflect the real-life clinical course. Therefore, the statistical comparison for the second treatment phase was adjusted for gender, etiology, age of epilepsy onset, and previously used ASMs. For refractory patients who required a rescue treatment phase with ASMs according to clinical needs, the study provided systematic follow-up in the context of the entire patient cohort. This enabled the evaluation of the cumulative response across all treatment phases by comparing different treatment sequences and conducting an intention-to-treat analysis as per the initial treatment allocation.

Other strengths of our study were its relatively large size and the representative sample of infants with IESS due to the PC-RCT design in the first treatment phase [29]. Another limitation is that the generalizability of the results may be limited by the single-center design. However, internal validity was optimized by the homogenous cohort and strict adherence to treatment protocols for KDT and ACTH at our center [29,43]. Although patients from our tertiary care center may have been selected for more severe cases, they were likely more prone to treatment failure and therefore represent an enriched population for the objectives of this follow-up study evaluating successive treatment phases. 

For infants who do not respond to standard initial treatments (ACTH, corticosteroids, and VGB), there is little evidence regarding optimal successive treatments. ASMs as second-line treatments reportedly show poor outcomes when compared with standard treatments [32,33,41]. Consequently, guidelines are mainly based on expert opinion [33]. KDT as a third-line treatment was reportedly effective in 9–35% of highly refractory cases of IESS [26,31,42], which was less effective than standard treatment but higher or similar to that seen in ASMs [31,32,33,41]. To the best of our knowledge, the present study is the first direct comparison of KDT versus ACTH as a second treatment for infants with IESS. Our study hypothesis was that KDT is equally effective as ACTH after failure of initial treatment.

After the first treatment phase, 66% of infants did not achieve sustained seizure freedom [29]. Of these, a total of 31% responded to the second treatment. This response is consistent with the literature: a multi-center prospective study of 118 infants with IESS (infantile spasm cohort) [41] showed a 37% response to a second treatment, and an RCT including refractory spasms showed a 44% response [31]. In our study, 15% of infants who did not respond to the second treatment phase required further treatment with ASMs and achieved sustained seizure freedom. The cumulative response after all treatment phases was 61%. This high overall response rate is likely due to the use of standardized protocols for the second treatment protocol in our study [41,44].

A comparison of different options for the second treatment phase showed sustained seizure freedom in 50% of infants who received second KDT after initial ACTH, while seizure freedom was achieved in 19% of infants who received second ACTH after initial KDT. Infants who received ASM as a second treatment responded in 9% of cases.

VGB was administered as prior monotherapy for only a short period of time before KDT or ACTH were added. Therefore, VGB was not considered as a stand-alone initial treatment in the analysis. However, we present a sub-analysis of treatments without or with VGB. Overall, no relevant differences were observed between the response to KDT and ACTH after all treatment phases without or with VGB. However, after the second treatment phase, sustained seizure freedom was more frequent for KDT (56%) compared to ACTH (0%).

The above-mentioned infantile spasm cohort [41] reported a higher response to a second standard treatment (55%) than to non-standard treatments (25%). Although this study included infants who received KDT alongside non-standard treatments, its results are not reported separately. In our study, KDT as a second treatment showed a higher response rate (50%) than expected for a non-standard treatment [41]. Currently, KDT is recommended for highly refractory children with all forms of epilepsy [45,46,47], including IESS [23,26,30,31,48,49], and has been shown as effective and safe in early childhood [31,43,45,46,47,48]. It has fewer severe adverse effects than hormonal treatment [29] and allows for adequate growth due to close nutritional monitoring [50], which we also confirm with this study.

Moreover, our results show that KDT administered early, as first or second treatment, is at least as effective as other standard treatments. The cumulative response after the first and second treatments was similar between the ACTH-KDT (68%) and the KDT-ACTH sequence (64%).

We therefore conclude that, regardless of whether KDT or ACTH is administered as a second treatment, the combination of treatments is necessary as rescue treatment, especially since hormonal treatment alone can only be used for a short period of time.

A small proportion of infants in our study were already eligible for epilepsy surgery during the first treatment. Sustained seizure freedom was achieved in 80% of these selected infants after epilepsy surgery. These results are consistent with recent literature data on appropriate candidacy for epilepsy surgery, which report seizure freedom rates between 48% and 80% [51,52].

In highly refractory infants requiring further treatment with ASMs, the response after a prior KDT-ACTH sequence was 38%, compared to 17% after ACTH-KDT. Although the overall response to the rescue treatment phase is consistent with the literature [26,30,31,32,33,41,42,49], to date, no analysis of different treatment sequences has been published. However, the cumulative response in the per-protocol groups after the first, second, and rescue treatment was largely similar for the ACTH-KDT-ASM (72%) and the KDT-ACTH-ASM sequence (78%). The intention-to-treat analysis per initial treatment allocation, which included infants after ASMs or epilepsy surgery as a second treatment, achieved a cumulative response of 60% after initial KDT and 63% after initial ACTH. We conclude that sustained seizure freedom is independent of the choice of initial treatment but can be maximized by a per-protocol second treatment (KDT or ACTH) instead of second ASMs.

We also performed a subgroup analysis stratifying infants according to concomitant use of VGB. VGB in combination with ACTH was shown to be more effective in terminating infantile epileptic spasms than hormonal treatment alone [9]. As previously reported for initial treatment, infants without VGB treatment showed no difference in sustained seizure freedom, whereas with concomitant VGB, the response to second KDT was better than to second ACTH. Infants receiving concomitant VGB formed a select group with longer epilepsy duration before first treatment with either KDT or ACTH [29] and potentially worse outcomes. However, these infants responded better to KDT than to ACTH during successive treatment phases. This confirms that KDT is highly effective for refractory epilepsy, including epileptic spasms [30,31,33,41]. The study was designed and approved in 2007, when guidelines recommended ACTH as a first-line treatment and reported ACTH as probably effective (Level I evidence), and probably more effective than VGB, which was considered the first choice only for IESS due to tuberous sclerosis (12). This was confirmed by guidelines from 2012 and 2015 [13,33]. To date, no Level I evidence has been reported for KDT as treatment for IESS [29]. We therefore conclude that KDT should be introduced at least as a second standard treatment for infantile epileptic spasms [12].

If corticosteroids are not possible due to poor general health, we also recommend starting KDT as early as possible, together with VGB as an initial combination [41,44]. However, the cumulative response after the first and second treatment phase, as well as after the first, second, and successive treatment phases, was largely similar for all treatment groups without or with prior VGB.

Our study confirms previous findings showing that the use of standardized protocols achieved the best seizure control [53,54], with ACTH being one of the best options of standard therapy [44]. Our study demonstrates that KDT as a second treatment is as effective as standard treatments [41], and that the effectiveness of KDT is increased in the presence of prior and concomitant VGB. We therefore recommend that KDT be considered as standard treatment for IESS and be implemented when hormonal treatment has failed.

## 5. Conclusions

In summary, we report a direct comparison of KDT and ACTH for the second treatment of infants with refractory IESS. Infants receiving second KDT after initial ACTH treatment showed a higher frequency of sustained seizure freedom than infants who received second ACTH treatment after initial KDT. Infants requiring further treatment with ASMs responded slightly better after a prior KDT-ACTH sequence than after an ACTH-KDT sequence. However, the cumulative frequency of sustained seizure freedom, after all treatment phases was similar for both initial treatment allocations. We therefore recommend that KDT be regarded as second treatment for IESS, and initiated once hormonal treatment has failed.

## Figures and Tables

**Figure 1 nutrients-17-02085-f001:**
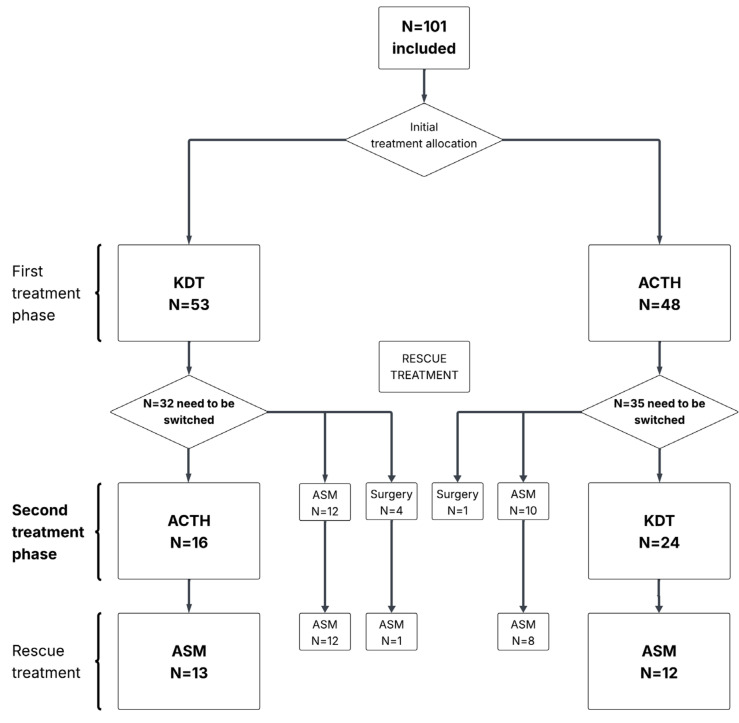
Flow of treatment sequences after switch from initial treatment. From the initial N = 101 patients, 32/53 had to be switched after the first KDT and 35/48 after the first ACTH. After the second treatment phase, N = 26 and N = 20 still needed further treatment. KDT: ketogenic diet, ACTH: adrenocorticotropic hormone, ASM: anti-seizure medication.

**Table 1 nutrients-17-02085-t001:** Characteristics of KDT.

Daily Nutritional Intake	
Calories (kcal, per kg/day) *	93.7 (63.6–166.7)
Protein (g, per kg/day) *	2 (1.3–3.6)
Lipids (g, per kg/day) *	8.9 (6–16.1)
Carbohydrates (g, per kg/day) *	1.2 (0.2–1.8)
Ratio (fat/non fat) *	2.8 (1.9–3.9)
Calories (kcal, per kg/day) *	93.7 (63.6–166.7)
Protein (g, per kg/day) *	2 (1.3–3.6)

* median, minimum–maximum.

**Table 2 nutrients-17-02085-t002:** Baseline patient characteristics per treatment arms of the second treatment phase.

SECOND TREATMENT	KDT-ACTH	ACTH-KDT	ASM	SURGERY	*p*-VALUE
N = 67	N = 16	N = 24	N = 22	N = 5	
Female *	9 (56%)	15 (63%)	13 (59%)	2 (40%)	0.827
Etiology known *	9 (56%)	17 (71%)	18 (82%)	5 (100%)	0.167
Age at epilepsy onset (months) **	4 (0–11.6)	2.4 (0–9.4)	3.7 (0–27)	2.9 (0.4–22)	0.413
Time from epilepsy onset to trial treatment (days) **	29 (7–631)	178 (7–412)	222 (7–792)	98 (69–857)	0.112
Number of ASMs before trial **	1.5 (0–5)	2 (0–6)	2 (0–7)	2 (1–5)	0.959
Concomitant Vigabatrin *	5 (31%)	19 (79%)	8 (36%)	2 (40%)	0.007
Psychomotor development age-appropriate at baseline *	3 (19%)	2 (8%)	3 (14%)	0 (0%)	0.628

* N (%), ** median, minimum–maximum. Differences between treatment groups were observed for (i) time from epilepsy onset to start of first treatment, and (ii) prior and concomitant Vigabatrin use.

**Table 3 nutrients-17-02085-t003:** Seizure freedom per treatment phase.

**FIRST TREATMENT PHASE**	**KDT**	**ACTH**		
	21/53 (40%)	13/48 (27%)		
**SECOND TREATMENT PHASE**	**ACTH**	**KDT**	**ASM**	**Epilepsy Surgery**
	3/16 (19%)	12/24 (50%)	2/22 (9%) *	4/5 (80%) **
**RESCUE TREATMENT**	**ASM**	**ASM**	**ASM**	**ASM**
	5/13 (38%)	2/12 (17%)	0/20 (0%)	0/1 (0%)

n/N (%). * 2/10 were seizure free after sequence ACTH-ASM; 0/12 after sequence KDT-ASM. ** 1/1 infant was seizure free after sequence ACTH-surgery, 3/4 infants were seizure free after sequence KDT-surgery.

**Table 4 nutrients-17-02085-t004:** Cumulative outcome at last follow-up observation per treatment sequence.

**FIRST TREATMENT PHASE**	**KDT**	**ACTH**
	21/53 (40%)	13/48 (27%)
**SECOND TREATMENT PHASE**	**KDT-ACTH**	**ACTH-KDT**
	24 */37 (64%)	25 **/37 (68%)
**RESCUE TREATMENT**	**KDT-ACTH-ASM**	**ACTH-KDT-ASM**
	29/37 (78%)	27/37 (72%)
**INTENTION-TO-TREAT *****	**Initial KDT**	**Initial ACTH**
	32/53 (60%)	30/48 (63%)

n/N (%). * N = 24 is the cumulative number of 21 infants responding to KDT in the first treatment phase, and N = 3 to ACTH in the second treatment phase. ** N = 25 is the cumulative number of 13 infants responding to ACTH in the first treatment phase, and N= 12 to KDT in the second treatment phase. *** Intention-to-treat shows response in all patients as per initial treatment allocation (including patients receiving rescue treatment with ASM (n = 22) or surgery (n = 5); as per Table 3).

**Table 5 nutrients-17-02085-t005:** Response (seizure freedom) to the second treatment phase comparing different treatment sequences.

Response to Second Treatment		Unadjusted OR (CI 95%)	*p*-Value	Adjusted OR (95% CI) *	*p*-Value
**ACTH-KDT** 12/24 (50%)	**KDT-ACTH** 3/16 (19%)	4.3 (0.98–19)	0.054	23.6 (2.3–237)	0.007
**ACTH-KDT** 12/24 (50%)	**ASM** 2/22 (9%)	10.0 (1.9–53)	0.007	66.1 (4–1088)	0.003
**ACTH-KDT** 12/24 (50%)	**SURGERY** 4/5 (80%)	0.25 (0.02–2.6)	0.244	0.05 (0.01–0.6)	0.067

* Adjusted for gender, etiology, age of epilepsy onset, and previously used ASMs. n/N (%), OR: odds ratio, CI: confidence interval.

## Data Availability

The original contributions presented in this study are included in the article/Appendix A. Further inquiries can be directed to the corresponding author.

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
