# Peer review of "Role of the Ketogenic Diet Therapy and ACTH as Second Treatments in Drug-Resistant Infantile Epileptic Spasms Syndrome"

_nutrients, 2025, doi:10.3390/nu17132085_

Round 1
Reviewer 1 Report
Comments and Suggestions for Authors
The article addresses a rare and clinically important problem: the efficacy of successive lines of treatment (ketogenic diet vs. ACTH) in infants with treatment-resistant infantile spasms syndrome (IESS). The comparison of consecutive therapies in this patient group is of particular importance and the data from the prospective study increases its value. To date, however, there have been few studies that have analysed KDT as a second-line therapy in such a systematic way.
The work is of considerable value in the field of paediatric neurology and nutritional therapy and represents a significant contribution to the existing literature.
The study protocol is transparent and well described.
A cohort of patients with refractory IESS was selected for this study.
A methodical approach to randomisation, observation and analysis of the results is essential.
A detailed statistical analysis was performed, including various methods such as odds ratio (OR), cumulative analysis and subgroup analysis.
The use of objective diagnostic tools, such as electroencephalography (EEG) and functional scales, is of paramount importance in this regard.
Although randomisation was performed in the initial phase, subsequent phases were not randomised for ethical reasons, which may lead to selection bias.
The lack of a full blind study may have implications for the evaluation of treatment effects, although the EEG evaluators were blinded.
It is important to note that the analysis of data from a single centre may limit the generalisability of the results.
The methodological quality of the study as well as ethical and logistical limitations were adequately considered and justified.
The results were presented in a structured, clear manner, with appropriate tables and statistical indicators.
The presentation of the data is commendable as it is both very good and understandable for clinical practitioners.
The authors rightly emphasise that:
KDT has been shown to be as effective as ACTH as a second-line treatment.
The combination of ACTH and KDT has been shown to be more effective than antiepileptic drugs (ASM) administered in isolation.
It is argued that KDT should be considered as a standard component of IESS therapy when hormonal treatment has not produced the desired result.
The conclusions drawn are consistent with previous findings and the existing literature, while providing new data on the course of therapy.
The conclusions drawn are reliable, data-driven and well motivated.
It is recommended that the description of limitations be expanded, with particular reference to the lack of randomisation in second- and third-line treatment and the recruitment of participants from a single centre.
The addenda should include additional data, such as a full summary of electroencephalogram (EEG) results and details of changes in pharmacotherapy during the study.
To avoid repetition and increase clarity, it is recommended to shorten or simplify the discussion.
Author Response
Response to Reviewer 1 Comments
|
||||||||||||||||||||||
1. Summary |
|
|
||||||||||||||||||||
Thank you very much for taking the time to review this manuscript. We are very pleased that you appreciate our manuscript and respond to your questions point per point. Please find the detailed responses below and the corresponding revisions/corrections highlighted/in track changes in the re-submitted files.
|
||||||||||||||||||||||
2. Questions for General Evaluation |
Reviewer’s Evaluation |
Response and Revisions |
||||||||||||||||||||
Does the introduction provide sufficient background and include all relevant references? |
Can be improved |
Please find the response in the point-by-point response letter. |
||||||||||||||||||||
Are all the cited references relevant to the research? |
|
|
||||||||||||||||||||
Is the research design appropriate? |
Can be improved |
|
||||||||||||||||||||
Are the methods adequately described? |
Can be improved |
|
||||||||||||||||||||
Are the results clearly presented? |
Can be improved |
|
||||||||||||||||||||
Are the conclusions supported by the results? |
stated as yes in the text |
|
||||||||||||||||||||
3. Point-by-point response to Comments and Suggestions for Authors |
||||||||||||||||||||||
Comments 1:
The article addresses a rare and clinically important problem: the efficacy of successive lines of treatment (ketogenic diet vs. ACTH) in infants with treatment-resistant infantile spasms syndrome (IESS). The comparison of consecutive therapies in this patient group is of particular importance and the data from the prospective study increases its value. To date, however, there have been few studies that have analysed KDT as a second-line therapy in such a systematic way. The work is of considerable value in the field of paediatric neurology and nutritional therapy and represents a significant contribution to the existing literature. The study protocol is transparent and well described. A cohort of patients with refractory IESS was selected for this study. A methodical approach to randomisation, observation and analysis of the results is essential. A detailed statistical analysis was performed, including various methods such as odds ratio (OR), cumulative analysis and subgroup analysis. The use of objective diagnostic tools, such as electroencephalography (EEG) and functional scales, is of paramount importance in this regard. Although randomisation was performed in the initial phase, subsequent phases were not randomised for ethical reasons, which may lead to selection bias. The lack of a full blind study may have implications for the evaluation of treatment effects, although the EEG evaluators were blinded. It is important to note that the analysis of data from a single centre may limit the generalisability of the results. The methodological quality of the study as well as ethical and logistical limitations were adequately considered and justified. The results were presented in a structured, clear manner, with appropriate tables and statistical indicators. The presentation of the data is commendable as it is both very good and understandable for clinical practitioners. The authors rightly emphasise that: KDT has been shown to be as effective as ACTH as a second-line treatment. The combination of ACTH and KDT has been shown to be more effective than antiepileptic drugs (ASM) administered in isolation. It is argued that KDT should be considered as a standard component of IESS therapy when hormonal treatment has not produced the desired result. The conclusions drawn are consistent with previous findings and the existing literature, while providing new data on the course of therapy.
The conclusions drawn are reliable, data-driven and well motivated. 1. It is recommended that the description of limitations be expanded, with particular reference to the lack of randomisation in second- and third-line treatment and the recruitment of participants from a single centre. 2. The addenda should include additional data, such as a full summary of electroencephalogram (EEG) results and details of changes in pharmacotherapy during the study. 3. To avoid repetition and increase clarity, it is recommended to shorten or simplify the discussion. |
||||||||||||||||||||||
Response to 1.) : Thank you for pointing this out. We agree with this comment. We have adjusted and shortened the discussion accordingly (page 9, paragraph 4, line 273 to page 12, paragraph 2, line 418).
Moreover , we have added and adjusted the following paragraph in the discussion (page 9, line 299-313: Given the available options available for the second treatment phase based on the existing literature [41, 26, 32, 33, 42, 31], assignment to the other protocol treatment (ACTH or KDT) was expected to be superior to ASMs. Randomisation at this time-point would not have been clinically and ethically appropriate which is a limitation to the comparison of treatments. Differences in patient characteristics between treatment arms, which may influence outcomes, result from selection due to failure of initial treatment and reflect the real-life clinical course. However, the statistical comparison for the second treatment phase was adjusted for gender, aetiology, age of epilepsy onset, and previously used ASMs.
Moreover, we have expanded the description of the limitations with particular emphasis to the lack of randomisation in the second- and third-line therapy and the recruitment from a single centre as follows: Page 5, paragraph 4, line 142-144: As the study was observational in the second treatment phase, no randomisation was performed. Treatment protocols and outcome assessments were identical for all treatment phases (29). Page 6, paragraph 1, line 168-169: As per protocol, infants who failed to respond were assigned to the second treatment phase (the respective other treatment), thus randomisation was not feasible.
Page 10, paragraph 2, line 316-319: |
||||||||||||||||||||||
Another limitation is, that generalisability of the results is limited by the single centre design. However, internal validity was optimised by the homogenous cohort and strict adherence to the treatment protocols for KDT and ACTH at our centre (43); (29).
|
||||||||||||||||||||||
Response to 2.) : We thank the reviewer for this comment and changed the text accordingly. We added a table on EEG data to the supplemental material (Supplemental Table S4)
Supplemental Table S4. EEG - Data n (%), * Hypsarrhyhtmia was present in all patients, although discontinuous or incipient in n= 10 infants at baseline. Epileptic discharges remained after 12 months in half of the patients and sleep spindels were present in around two thirds.
And we added the following sentence to the text: Page 9, paragraph 2, line 267-270: EEG-Data are displayed in in Supplemental Table S4. Hypsarrhyhtmia was present in all patients, although discontinuous or incipient in n= 10 infants at baseline. Epileptic discharges were still present in half of the patients after 12 months, and sleep spindels were present in around two thirds.
|
||||||||||||||||||||||
4. Response to Comments on the Quality of English Language |
||||||||||||||||||||||
Point 1: |
||||||||||||||||||||||
Response 1: No changes recommended by reviewer 1. |
||||||||||||||||||||||
5. Additional clarifications. None |
||||||||||||||||||||||
|

Reviewer 2 Report
Comments and Suggestions for Authors
While the ketogenic diet (KDT) is a nutritional therapy, the manuscript is primarily centered around neurological treatment efficacy and seizure outcomes. To better align with Nutrients, I recommend:
- The abstract and introduction would benefit from professional English language editing to improve clarity and readability.
- Line 53 and throughout the manuscript: Please unify the reference list, ensuring that citation formatting is consistent with Nutrients
- Expanding the discussion of nutritional implications of KDT (e.g., effects on growth, metabolism, tolerance, side effects).
- Outcome Measures – Nutritional Data Missing, anthropometric changes (weight/height percentiles).
Author Response
Response to Reviewer 2 Comments
|
|||||||||||||||||||||||||||||||||||||||||||||||||||||||||||||||||||||||||||||||||||||
1. Summary |
|
|
|||||||||||||||||||||||||||||||||||||||||||||||||||||||||||||||||||||||||||||||||||
Thank you very much for taking the time to review this manuscript. We are very pleased that you appreciate our manuscript and respond to your questions point per point. Please find the detailed responses below and the corresponding revisions/corrections highlighted/in track changes in the re-submitted files.
|
|||||||||||||||||||||||||||||||||||||||||||||||||||||||||||||||||||||||||||||||||||||
2. Questions for General Evaluation |
Reviewer’s Evaluation |
Response and Revisions |
|||||||||||||||||||||||||||||||||||||||||||||||||||||||||||||||||||||||||||||||||||
Does the introduction provide sufficient background and include all relevant references? |
Must be improved |
Please find the response in the point-by-point response letter. |
|||||||||||||||||||||||||||||||||||||||||||||||||||||||||||||||||||||||||||||||||||
Are all the cited references relevant to the research? |
|
|
|||||||||||||||||||||||||||||||||||||||||||||||||||||||||||||||||||||||||||||||||||
Is the research design appropriate? |
Must be improved |
|
|||||||||||||||||||||||||||||||||||||||||||||||||||||||||||||||||||||||||||||||||||
Are the methods adequately described? |
Can be improved |
|
|||||||||||||||||||||||||||||||||||||||||||||||||||||||||||||||||||||||||||||||||||
Are the results clearly presented? |
Can be improved |
|
|||||||||||||||||||||||||||||||||||||||||||||||||||||||||||||||||||||||||||||||||||
Are the conclusions supported by the results? |
Must be improved |
|
|||||||||||||||||||||||||||||||||||||||||||||||||||||||||||||||||||||||||||||||||||
3. Point-by-point response to Comments and Suggestions for Authors |
|||||||||||||||||||||||||||||||||||||||||||||||||||||||||||||||||||||||||||||||||||||
While the ketogenic diet (KDT) is a nutritional therapy, the manuscript is primarily centered around neurological treatment efficacy and seizure outcomes. To better align with Nutrients, I recommend:
Response to 1.) : Thank you for pointing this out. It is very important to highlight the role of ketogenic diet therapy (KDT) in this context, as waiting lists for KDT are quite long and specialist tend to offer anti-seizure medications first even if KDT is effective and safe and feasible in early childhood. With our data, we want to emphasise the role of KDT in the early treatment course. We have adjusted the abstract and the introduction as follows (changes are marked in the text with track changes):
Page 1, paragraph 1, line 17-35 Abstract: Objectives: Aim of this study was to evaluate the effectiveness of sequential treatments with adrenocorticotropic hormone (ACTH) or ketogenic diet therapy (KDT) in infants with infantile epileptic spasms syndrome (IESS) who did not achieve seizure freedom after initial treatment with either KDT or ACTH. Methods: We conducted a cohort study following a parallel-cohort randomised controlled trial comparing KDT with ACTH as first-line treatment for IESS. Infants who failed to respond were switched per protocol to the alternative treatment (ACTH or KDT), or, if this was not feasible or unsuccessful, to anti-seizure medications (ASMs). The primary outcome was the frequency of sustained seizure freedom at last follow-up. Results: Of 101 infants allocated to the initial treatment phase, n=67 required further treatment. Of these, 31% (21/67) achieved sustained seizure freedom after the second treatment phase and 15% (7/46) after rescue treatment with ASMs. KDT as second treatment achieved sustained seizure freedom in 50% (12/24) compared to 19% (3/16) with ACTH and 9% (2/22) with ASMs. The cumulative response rate after first and second treatment was 64% for the KDT-ACTH sequence and 68% for the ACTH-KDT sequence. The cumulative response after all three treatment phases was 78% (KDT-ACTH-ASM) and 72% (ACTH-KDT-ASM), respectively. Conclusions: KDT is at least as effective as ACTH as second treatment and significantly more effective than ASMs in achieving sustained seizure freedom in infants with IESS.
Moreover, we have adapted the introduction as follows (changes are marked in the text with track changes): Page 2, paragraph 1, line 40 to Page 3, paragraph 2, line 94)
Infantile Epileptic Spasm Syndrome (IESS) occurs in approximately 22 out of 100.000 live-born infants (1), accounting for approximately 10% of all epilepsies in the first years of life (2, 3). IESS was proposed by the International League Against Epilepsy (ILAE) as a new terminology to cover the clinical spectrum of all cases of epileptic spasms, even when hypsarrhythmia in the EEG and developmental delay are not or not yet present (4). The syndrome may be preceded by Early Infantile Developmental Epileptic Encephalopathies (EIDEE), which exhibit the clinical phenotype of IESS at approximately 3 to 4 months of age (5). Early seizure control and resolution of hysparrhythmia are considered to be positive determinants of developmental outcome (6), therefore, timely, effective, and safe treatment is essential. The longer hypsarrhythmia and epileptic spasms persist, the more delayed age-appropriate development becomes (7, 8, 9), and approximately 70% of children develop drug-resistant seizures with intellectual disability (10). First-line therapies such as adrenocorticotropic hormone (ACTH), Vigabatrin (VGB), and oral steroids have shown to be effective (10, 11, 12, 13, 14), but have high relapse rates (15); (16) and severe adverse effects with ACTH (12, 17) and VGB (18, 19, (20). Recently, the combination of ACTH and VGB has been reported to be more effective than hormonal treatment alone in terminating infantile epileptic spasms (9). Ketogenic diet therapy (KDT) has shown substantial effectiveness (21, 22, 23, 24, 25, 26, 27, 28, 29, 30, 31) for refractory epileptic spasms after failure of standard treatment with hormonal therapy or VGB. Most studies were retrospective, but four prospective studies reported the use of KDT in IESS (29, 23, 30, 31). Hong and co-workers showed that 37% of infants became seizure free, but 71% had been pre-treated with corticosteroids and VGB (23). Our study group recently published the results of a prospec-tive parallel cohort randomized controlled trial (PC-RCT) (29). The study showed that KDT as initial treatment was as effective as ACTH in achieving long-term seizure freedom, but was better tolerated. Sharma et al. reported on the use of the Modified Atkins Diet (MAD) for the treatment treating of epileptic spasms refractory to hormonal treatment and randomised to either KDT or ASM. Their study showed that KDT was superior to the addition of a further ASM (23% vs. 0%) (30). In a recent study, Schoeler et al. randomised infants under two years of age with drug-resistant epilepsy to either KDT or ASM and demonstrated comparable efficacy (seizure-freedom in 11% vs. 13%, respectively), as well as a better quality of life in the KDT group (31). This cohort included 60% of refractory IESS. In infants with IESS who do not respond to standard treatments, there is sparse evidence for optimal successive treatments (32). Therefore, guidelines are primarily based on expert opinion (32, 33). In highly refractory IESS cases, response rates range from 9 to 44% (26, 34, 31), which is lower than with standard treatment, but higher than with some ASMs (14); (31, 32, 30). However, updated guidelines for the German Speaking Countries and data from the KIWE trial support the use of KDT immediately after failure of hormonal treatments and VGB (32, 31). The present manuscript reports on the follow-up of our prospective PC-RCT in infants with IESS (29) who failed initial treatment with either KDT or ACTH. The aim of this study was to evaluate the effectiveness of successive treatments. Specific objectives were to assess sustained seizure freedom until the last follow-up visit: i. in response to ACTH or KDT as second treatment; ii. in response to ASM as second treatment (rescue treatment); iii. in response to ASM as third treatment (KDT-ACTH-ASM or ACTH-KDT-ASM); and iv. to determine the overall most effective treatment sequence. Our hypothesis was that KDT is at least as effective as ACTH as second treatment for refractory IESS.
|
|||||||||||||||||||||||||||||||||||||||||||||||||||||||||||||||||||||||||||||||||||||
Response to 2.) : We adjusted the citations accordingly. We thank the reviewer for this comment and changed the text accordingly.
Response to 3.) and 4.): We have added data on nutritional outcome and adverse affects of the diet as follows: Page 6, paragraph 2 and 3, lines 182-188 and added Table 1, line 185:
Table 1 displays nutritional characteristics of the ketogenic diet prescription. Table 1. Characteristics of KDT
*median, minimum-maximum.
And we have added the following data the text and Tables S5 and S6 to the supplemental material:
Page 9, paragraph 3, line 271-272.
Adverse effects of KDT are shown in Supplemental Table S5, and growth data in Supplemental Table S6.
See also supplemental material for Supplemental Tables S5 and S6
Supplemental Table S5. Adverse effects of KDT *n (%). KDT was administered in a total of n= 56 patients (including 32 patients of the first treatment with KDT who needed second treatment and 24 patients who received KDT as second treatment).
Supplemental Table S6. Z-scores of Growth during KDT *median (minimum and maximum). No major changes in growth z-scores were observed. Infants remained on their percentiles, lower in single cases due to the underlying aetiology (metabolic epileptic encephalopathies) and limited mobility, which we recently published (Maass et al. 2024). Moreover, we have changed the discussion accordingly, as follows: Page 11, paragraph 1, line 360-368. KDT is recommended for highly refractory children with all forms of epilepsy [45, 46, 47], including IESS [23, 26, 48, 30, 49, 31] and has been shown as effective and safe in early childhood [31, 43, 45, 46, 47, 48]. It has fewer severe adverse effects than hormonal treatment [29], and allows for adequate growth due to close nutritional monitoring [50], which we confirm also with this study. Moreover, our results show that KDT administered early, as first or as second treatment, is at least as effective as other standard treatments. The cumulative response after first and second treatment was similar between the ACTH-KDT sequence (68%) and the KDT-ACTH sequence (64%).
|
|||||||||||||||||||||||||||||||||||||||||||||||||||||||||||||||||||||||||||||||||||||
4. Response to Comments on the Quality of English Language |
|||||||||||||||||||||||||||||||||||||||||||||||||||||||||||||||||||||||||||||||||||||
Point 1: |
|||||||||||||||||||||||||||||||||||||||||||||||||||||||||||||||||||||||||||||||||||||
Response 1: Changes are highlighted in the text. |
|||||||||||||||||||||||||||||||||||||||||||||||||||||||||||||||||||||||||||||||||||||
5. Additional clarifications |
|||||||||||||||||||||||||||||||||||||||||||||||||||||||||||||||||||||||||||||||||||||
A review of the English language was performed by the Institute of Linguistics, University of Vienna, Austria.
|

Round 2
Reviewer 2 Report
Comments and Suggestions for Authors
The revised manuscript provides important insights into second-line treatment options for infantile epileptic spasms syndrome (IESS), with a well-designed prospective cohort follow-up to a previous RCT. The research question is clinically relevant, and the conclusions are largely supported by the data. Overall, with thorough language revision and minor structural tightening, this manuscript would make a strong contribution to the field.